# Preparation of Sr_2_CeZrO_6_ Refractory and Its Interaction with TiAl Alloy

**DOI:** 10.3390/ma16237298

**Published:** 2023-11-23

**Authors:** Fuli Bian, Zheyu Cai, Jian Liu, Yu Liu, Man Zhang, Yixin Fu, Kailiang Zhu, Guangyao Chen, Chonghe Li

**Affiliations:** 1Fire Research Institute of Shanghai of MEM, Shanghai 200032, China; bianfuli@shfri.cn (F.B.); zhukailiang@shfri.cn (K.Z.); 2State Key Laboratory of Advanced Special Steel & Shanghai Key Laboratory of Advanced Ferrometallurgy & School of Materials Science and Engineering, Shanghai University, Shanghai 200444, China; zycai2022@shu.edu.cn (Z.C.); liuzuozuo@shu.edu.cn (J.L.); liuyucumt@shu.edu.cn (Y.L.); manzhang@shu.edu.cn (M.Z.); fyx13194382608@shu.edu.cn (Y.F.); 3Shanghai Special Casting Engineering Technology Research Center, Shanghai 201605, China; 4Zhejiang Institute of Advanced Materials, Shanghai University, Jiaxing 314100, China

**Keywords:** Sr_2_CeZrO_6_, refractory, induction melting, TiAl alloys, interaction

## Abstract

Vacuum induction melting in a refractory crucible is an economical method to produce TiAl-based alloys, aiming to reduce the preparation cost. In this paper, a Sr_2_CeZrO_6_ refractory was synthesized by a solid-state reaction method using SrCO_3_, CeO_2_ and ZrO_2_ as raw materials, and its interaction with TiAl alloy melt was investigated. The results showed that a single-phase Sr_2_CeZrO_6_ refractory could be fabricated at 1400 °C for 12 h, and its space group was Pnma with a = 5.9742(3) Å, b = 8.3910(5) Å and c = 5.9069(5) Å. An interaction layer with a 40μm thickness and dense structure could be observed in Sr_2_CeZrO_6_ crucible after melting TiAl alloy. Additionally, the interaction mechanism showed that the Sr_2_CeZrO_6_ refractory dissolved in the alloy melt, resulting in the generation of Sr_3_Zr_2_O_7_, SrAl_2_O_4_ and CeO_2−x_, which attached to the surface of the crucible.

## 1. Introduction

TiAl alloys have been considered as novel lightweight construction materials due to their low density (approximately 4.0 g/cm^3^) and high specific strength [1]. They have the potential to replace heavier Ni-based superalloys, resulting in weight reduction and enhancing the thrust-to-weight ratio [2,3]. Although TiAl alloys have substantial performance advantages, their elevated production costs are a factor limiting their further development [4,5].

Investment casting technology with induction skull melting (ISM) has been the commercial method for the manufacture of TiAl alloy parts [6]. However, the implementation of ISM significantly influences the ultimate cost of the casting products, which has contributed to the elevated levels of casting rejections. This is because this method does not easily achieve suitable superheating. In order to solve this problem, a suitable preheat temperature for the casting mold should be used. This results in the occurrence of the melt–mold interaction, and solidification defects are thus also introduced. Vacuum induction melting (VIM) of TiAl alloys using the refractory crucibles offers a viable approach to attaining optimal superheating and reducing the production expenses. However, the key problems are the selection of exceptionally stable and cost-effective refractory in crucible manufacturing [7,8,9,10].

Until now, researchers have explored various oxide materials as refractory crucibles for melting TiAl alloys, such as Y_2_O_3_ [11], CaO [12], ZrO_2_ [13] and Al_2_O_3_ [14]. However, these refractory materials applied for melting TiAl alloy also have shortcomings [15]. Toshimitsu demonstrated the feasibility of melting TiAl alloys in crucibles made of Y_2_O_3_, ZrO_2_, and Al_2_O_3_ [13]. Notably, the oxygen levels in the TiAl alloy were relatively low (0.12 wt.%) using the Y_2_O_3_ crucible. In contrast, the absorption of oxygen from the ZrO_2_ and Al_2_O_3_ crucibles occurred at a notably higher level (0.96 and 1.57 wt.%, respectively). Koichi’s work presented that the oxygen concentration could be controlled at 0.1–0.13 wt.% with a holding smelting time ranging from 5 to 20 min in the CaO crucible [16]. From a thermodynamic perspective, Y_2_O_3_ and CaO had higher stability than that of Al_2_O_3_ and ZrO_2_. Thus, they were more suitable for melting TiAl alloys. However, there are significant challenges that need to be addressed before industrial-scale manufacture. For instance, Y_2_O_3_ has an inherent drawback of poor thermal shock resistance, and CaO exhibits a hygroscopic nature [17,18]. Given these considerations, it becomes imperative to explore and develop a new stable refractory for the melting of TiAl alloys.

Novel alkaline earth zirconate materials, such as Sr-Zr oxides, exhibit the essential properties for melting titanium alloys. For example, SrZrO_3_ was an attractive candidate material due to its exceptional resistance to corrosion, especially in alkaline melts and vapors [19]. Generally, compounds with perovskite ABX_3_ structure are very well-known inorganic materials. The A and B sites in perovskite materials possess the capability to accommodate diverse metal cations, providing an avenue to regulate both the chemical compositions and the properties of perovskite materials. Substituting elements on the B sites can lead to the formation of double perovskite A_2_B′B″O_6_ compounds, where A represents an alkaline-earth metal, and B′ and B″ represent two different transition metal elements. The positioning of B′ and B″ cations within the crystal structure can vary, either occupying indistinguishable sites or distinct sites, contingent upon their charge and ionic radii. Double perovskites exhibit a wide array of intriguing properties owing to their diverse compositions and structures, prompting extensive investigations into their structural characteristics [20]. In order to develop an Sr-Zr series perovskite oxide refractory for melting titanium alloys, our group attempted to prepare a novel refractory by doping rare earth elements, followed by evaluating their stability for melting titanium alloys [21]. The rare oxide such as CeO_2_ has a high melting point (nearly 1950 °C). It is typically stable at high temperatures, making it potentially useful in high-temperature applications, such as in high-temperature lubricants or coating materials [22]. In this study, CeO_2_ was introduced into the preparation of the Sr-Zr oxide refractory. It could be considered that the Ce dopant could replace the position of Zr elements in the Sr-Zr oxides. Thus, a double perovskite structure Sr_2_CeZrO_6_ refractory was fabricated, and then its interaction with melting TiAl alloys was investigated. Until now, there have not been any prior exploration into utilizing Sr_2_CeZrO_6_ refractory for preparing TiAl alloys.

In this paper, firstly, the manufacturing process involved the synthesis of a Sr_2_CeZrO_6_ refractory through a solid-state approach utilizing industrial-grade SrCO_3_, CeO_2_ and ZrO_2_ raw materials. Subsequently, the formation of the Sr_2_CeZrO_6_ crucible was achieved via shaping and sintering. Utilizing XRD and SEM, the synthesized powders were analyzed to ascertain the phase composition and microstructure. The structure of Sr_2_CeZrO_6_ refractory was investigated through the Rietveld method. Subsequently, TiAl alloy was melted in the Sr_2_CeZrO_6_ crucible, facilitating an exploration into the interaction between the TiAl alloy and Sr_2_CeZrO_6_ crucible, along with an investigation into the corresponding interaction mechanism.

## 2. Experiment

SrCO_3_ (99.9%), ZrO_2_ (99.9%) and CeO_2_ (99.9%) raw materials were employed for the preparation of a SrCeZrO_6_ refractory according to the solid-state reaction. The XRD (Bruker GADDS, Cambridge, MA, USA) patterns for the raw materials are shown in Figure 1. The raw materials were accurately measured following a mole ratio of n(SrCO_3_): n(ZrO_2_): n(CeO_2_) = 2:1:1. Subsequently, they were subjected to ball milling (MD-2 L, Nanjing, China), followed by drying at 120 °C for 12 h. Then, the discs with φ20 mm × 3 mm were fabricated at 120 MPa, holding for 2 min. Finally, the unfired discs were heat-treated at 1400 °C for 12 h.

Prior to the phase analysis, the sintered discs were ground into powders and then sieved through a 400-mesh sieve. XRD was used to analyze the phase structure of the powders through Rietveld refinement via GSAS-II software (https://subversion.xray.aps.anl.gov/trac/pyGSAS). Microstructure analysis was carried out using scanning electron microscopy (FEI Nova nano SEM450, Sydney, Austria).

The Sr_2_CeZrO_6_ refractory material underwent cold isostatic pressing to form crucibles using a U-shaped steel mandrel measuring 3.5 cm in width and 4.5 cm in height. in this shaping process, a pressure of 120 MPa was applied for a duration of 3 min. Then, the crucible biscuits were sintered at 1700 °C and held for 4 h. The slow heating rate of 2 °C/min was carefully controlled to prevent the formation of cracks.

Our goal was to assess the interface stability between the Sr_2_CeZrO_6_ refractory and the TiAl alloy melt. Before the melting experiment, a TiAl master alloy with an equimolar ratio was prepared using Al pellets (>99.99%) and sponge Ti (>99.9%) in a water-cooled copper crucible. In the VIM furnace, the space surrounding the crucible was filled with Al_2_O_3_ ramming mass. Then, the master alloy was inserted into the crucible. The furnace chamber was evacuated to 10^−3^ mbar. The high-purity argon gas was backfilled into the chamber at least three times. This effectively prevented oxygen contamination during the melting. As the molten alloy became visible, high-purity argon gas was reintroduced, and the temperature was slowly raised to 1600 °C and maintained for 5 min. Then, the alloy melt was allowed to cool in the crucible. X-ray diffraction was used to analyze the crucible surface in order to examine the interaction products between them. The interaction behavior was investigated after the fabrication of the samples using a buzz saw. Selected sections underwent examination using a digital microscope (VHX-1000). Interaction analyses for the microstructure were conducted using a scanning electron microscope with an energy dispersive spectrometer (EDS).

## 3. Results and Discussion

### 3.1. Synthesis of Sr_2_CeZrO_6_ Powder

Figure 2 shows the powder synthesized using SrCO_3_, CeO_2_ and ZrO_2_ after sintering at 1400 °C for 12 h. The sintered powder exhibited a significant sintering phenomenon, as shown in Figure 2a. After the crushing process, the edges of the powder were distinct, indicating that this synthesized condition was sufficient for preparing this refractory. Figure 2b depicts the backscatter electron (BSE) image from Figure 2a. It can be seen that the synthesis degree of this compound was good. Furthermore, chemical element mapping of Sr, Ce, Zr, and O was carried out (Figure 2b–f). The uniform distribution of these elements is evident, showing no signs of accumulation or segregation.

To further confirm the crystal structure of Sr_2_CeZrO_6_, the XRD pattern of the sintered powder was refined by the Rietveld method. The Rietveld refinement of the XRD data of the powder in the 2θ° angle 10~120° is shown in Figure 3. Indexing of the XRD spectra by Jade 6.0 program revealed that Sr_2_CeZrO_6_ had a similar peak pattern to SrCeO_3_ (Pnma 62). Therefore, the XRD data of the Sr_2_CeZrO_6_ sample were refined using the orthogonal structure Pnma space group. In this structural model, Sr atoms occupied the 4c position at coordinates (0.45, 025, 0), while both Ce and Zr atoms occupied the 4a position (0, 0, 0) with an equal occupancy of 0.5:0.5. O atoms were positioned at 4c (0, 0.25, 0) and 8d (0, 0, 0.2). Figure 3 illustrates the refined structural model of the Sr_2_CeZrO_6_. The experimental intensity (depicted by the green dotted line) of the Sr_2_CeZrO_6_ coincided with the simulated intensity (shown as the red continuous line). The lattice constants were a = 5.9742(3) Å, b = 8.3910(5) Å, c = 5.9069(5) Å, respectively, with a unit cell volume of 296.11(6) Å3 using the least-squares method. The reliability factors Rwp and GOF stood at 8.94% and 1.57, respectively. This indicated that the Rietveld refinements were conducted at a reasonably high standard. Table 1 outlines the refined structural parameters of the Sr_2_CeZrO_6_.

### 3.2. Phase Constitution of Sr_2_CeZrO_6_ Crucible

The SEM picture of the Sr_2_CeZrO_6_ crucible surface after sintering at 1700 °C and holding for 4 h is shown in Figure 4. The backscatter electron (BSE) picture in Figure 4a showed that the crucible consisted of a single phase. The sintering process of the crucible was undertaken in a silicon molybdenum rod furnace with an air atmosphere. During the prolonged calcination process, substances inside the furnace lining could volatilize and deposit on the surface of the crucible, resulting in the formation of some impurities. Table 2 indicates the EDS results of spots 1 and 2. It can be seen that the grains in the crucible surface consisted of Sr, Ce, Zr and O elements, which was consistent with the theoretical ratio. The XRD pattern in Figure 4b shows that only the Sr_2_CeZrO_6_ phase could be detected on the crucible surface, confirming the analysis in Figure 4a.

### 3.3. Interfacial Interaction

The macroscopic picture of the cooled alloy and the crucible is shown in Figure 5. Evidently, the alloy was separated from the crucible matrix. In the process of the melting, the alloy with its high chemical activity was able to permeate into the crucible refractory, resulting in the generation of a black area (contact layer). Due to the rapid heating rate during the melting process, some cracks were generated. In the future, decreasing the heating rate should be considered in order to reduce the occurrence of cracks. Additionally, the integrity of the crucible was not changed. The detailed analysis is described below.

The macroscopic pictures of the cross-sections of the crucibles before and after melting are shown in Figure 6. Before the melting, the crucible displayed a glossy yellow sheen, and the inner wall of the crucible was flat, as shown in Figure 6a. From Figure 6b, it can be seen that the crucible exhibited a black-gray color. Additionally, there was no significant erosion layer observed.

Figure 7 shows the microstructure of the crucible surface before and after the melting. As shown in Figure 7a, a smooth surface could be observed before the melting, and the boundaries were obvious. However, after the melting, the crucible surface appeared uneven, and the grain boundaries had disappeared. Evidently, during the melting, the crucible surface corroded. Figure 7c shows the magnified picture of area A in Figure 7b. A combination of block and strip grains appeared on the crucible surface, as shown in Figure 7c. Table 3 exhibited that the grains with the block shapes consisted of Sr, Zr, Ce and O elements. The XRD pattern for the crucible surface in Figure 7d shows that it consisted of Sr_3_Zr_2_O_7_, CeO_2−x_ and SrAl_2_O_4_ phases. This can be elucidated in conjunction with the suggested interaction model outlined below.

The SEM pictures of the cross-section of the crucibles before and after melting are shown in Figure 8. As shown in Figure 8a, the inner wall of the crucible was flat, and some pores were observed in the crucible matrix. EDS results in Table 4 show that the crucible matrix (spots 8 and 9) consisted of an Sr_2_CeZrO_6_ phase. After melting, an interaction layer (~40 μm thickness) and dense structure could be observed in the crucible, as shown in Figure 8b. EDS results indicated that the crucible matrix (spot 10) exhibited a different elementary composition from the interaction layer (spots 11 and 12). As shown in Figure 8d–f, there was a large number of Al elements, which were enriched in the interaction layer. Additionally, the Zr elements exhibited a clear downward trend in the interaction layer in comparison with those in the crucible matrix. It can be concluded that a movement of Al and Zr elements occurred during the interaction between them. Actually, because of the similar structures of the Zr and Ti elements, both of them exhibited excellent compatibility, resulting in the easy dissolution of Zr elements into the alloy melt. From the analysis in Figure 7, it can be seen that the Al reacted with the decomposed refractory to generate SrAl_2_O_4_, resulting in the enrichment of Al elements in the interaction layer. Additionally, there were essentially no Ti elements, which were residual along the crucible sidewall. This indicated that this refractory exhibited a good non-wettability during the alloy melts, consistent with the analysis in Figure 7c.

The detailed electronic states of the constituent elements in the interaction layer were characterized by X-ray photoelectron spectroscopy (XPS) measurements, as shown in Figure 9. It can be seen that the valence states of Sr and Zr were both in the positive tetravalent state, corresponding to the valence states in SrTiO_3_ and ZrO_2_, respectively. However, the coexistence of Ce^4+^ and Ce^3+^ was observed in this layer. This indicates that there was non-stoichiometric CeO_2_ in the interaction layer (i.e., CeO_2−x_). Evidently, the interaction between the refractory and the alloy melt caused the phase change in the interaction layer. The detailed interaction mechanism is described next.

Figure 10 illustrates the dependence of ΔG0 vs. the temperature for the formation of TiO_2_, TiO, Al_2_O_3_, CeO, ZrO_2_, SrO and Sr_3_Zr_2_O_7_, according to results obtained from the HSC software (Version 6.1). The Gibbs free energy of Sr_2_CeZrO_6_ was not confirmed. Because no interaction products, such as TiO_2_ and TiO, could be detected (see Figure 7d), it can be concluded that the interaction mechanism between them was still the dissolution of the crucible refractory in the alloy melt. The dissolution Equation (1) is described below:(1)2Sr2CeZrO6+Ti→in meltSrO+Sr3Zr2O7+2CeO2

From Figure 10, it can be seen that the decomposed product of Sr_3_Zr_2_O_7_ exhibited a better stability than that of TiO_2_ and TiO. Thus, due to the high chemical activity of the TiAl alloy melt, it further dissolved in the alloy melt, leading to the generate of SrO and the solution of Zr and O elements into the alloy melt. Additionally, CeO_2_ exhibited a worse stability than that of TiO. It reacted with the alloy melt, resulting in the generation of TiO. However, this reaction occurred along the solid–liquid interface, and the Ti melt exhibited a relatively high solubility with the O elements. Thus, no TiO could be detected in the crucible. The Equations (2) and (3) were showed as follows.
(2)Sr3Zr2O7→in meltSrO+Zr+O 
(3)CeO2+Ti→in meltTiO+CeO2−x

As the O elements dispersed along the interface between the crucible and the alloy melt, it further combined with the Al elements to generate Al_2_O_3_ products. Then, Al_2_O_3_ reacted with SrO, resulting in the generation of SrAl_2_O_4_. The reaction Equations (4) and (5) are presented below.
(4)O+Al→in meltAl2O3 
(5)SrO+Al2O3→in meltSrAl2O4

## 4. Conclusions

In this study, a novel perovskite SrCeZrO_6_ refractory was synthesized using the solid-state method. The interaction of the SrCeZrO_6_ refractory with a TiAl alloy melt was investigated. Conclusions were obtained as follows: (1)The pure compound Sr_2_CeZrO_6_ was successfully synthesized by a solid-state reaction after sintering at 1400 °C for 12 h. The Sr_2_CeZrO_6_ phase showed Pnma space group symmetry with a = 5.9742(3) Å, b = 8.3910(5) Å, c = 5.9069(5) Å.(2)The Sr_2_CeZrO_6_ refractory crucible maintained an intact shape after contacting with the TiAl alloy melt, and the thickness of the interaction layer was about 40 μm.(3)The Sr_2_CeZrO_6_ refractory decomposed into Sr_3_Zr_2_O_7_, SrAl_2_O_4_ and CeO_2−x_. The products were generated by the dissolution of the Sr_2_CeZrO_6_ refractory in the alloy melt, which was the main factor responsible for the interaction mechanism between them. This study provides some theoretical guidance for the development of new refractory materials for melting titanium alloys in the future.

## Figures and Tables

**Figure 1 materials-16-07298-f001:**
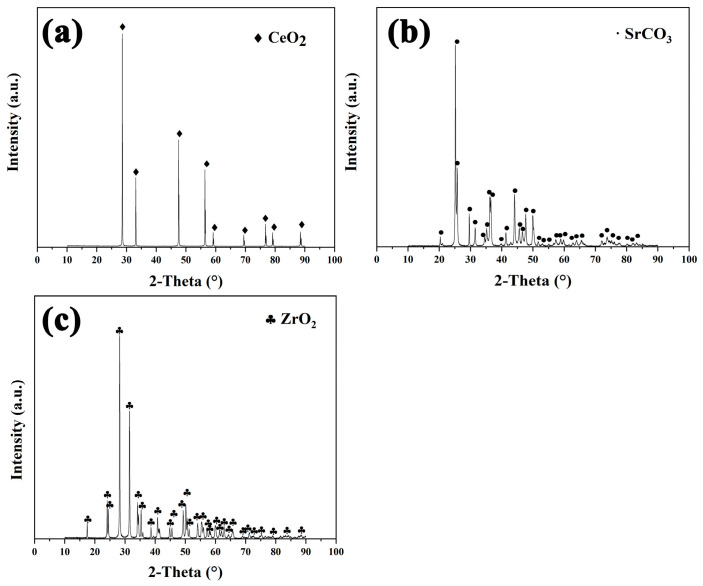
XRD patterns for the raw materials. (**a**) CeO_2_; (**b**) SrCO_3_; (**c**) ZrO_2_.

**Figure 2 materials-16-07298-f002:**
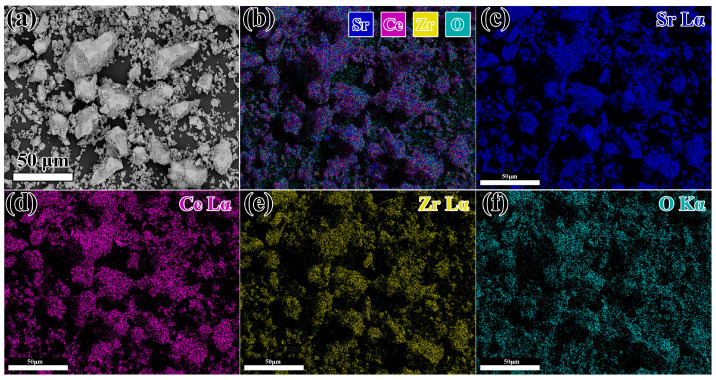
(**a**) The SEM pictures of synthesized powder after maintenance at 1400 °C for 12 h; (**b**–**f**) EDS elemental mapping pictures of Sr Lα, Ce Lα, Zr Lα and O Kα.

**Figure 3 materials-16-07298-f003:**
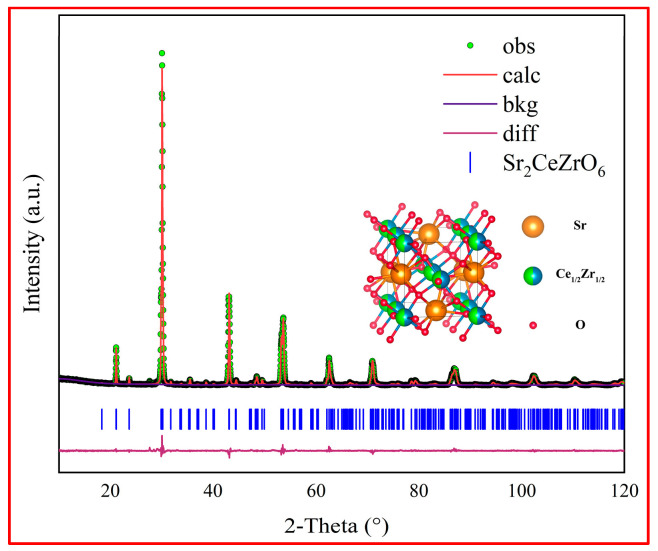
Experimentally observed (dots), Rietveld-calculated (continuous line), and difference (continuous bottom line) profiles for Sr_2_CeZrO_6_ after Rietveld analysis for the XRD data using the Pnma space group. The vertical tick marks above the difference plot showed the Bragg peak positions.

**Figure 4 materials-16-07298-f004:**
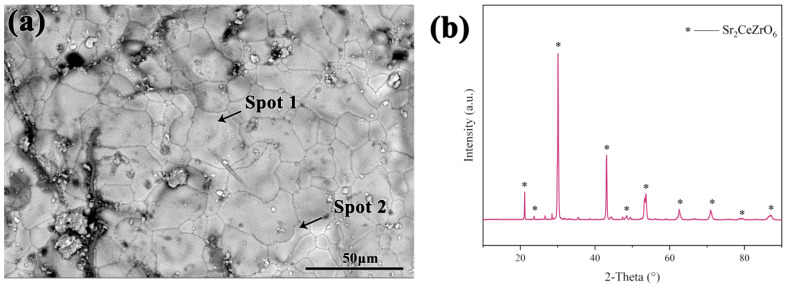
(**a**) The SEM picture of Sr_2_CeZrO_6_ crucible surface; (**b**) XRD pattern of the crucible surface.

**Figure 5 materials-16-07298-f005:**
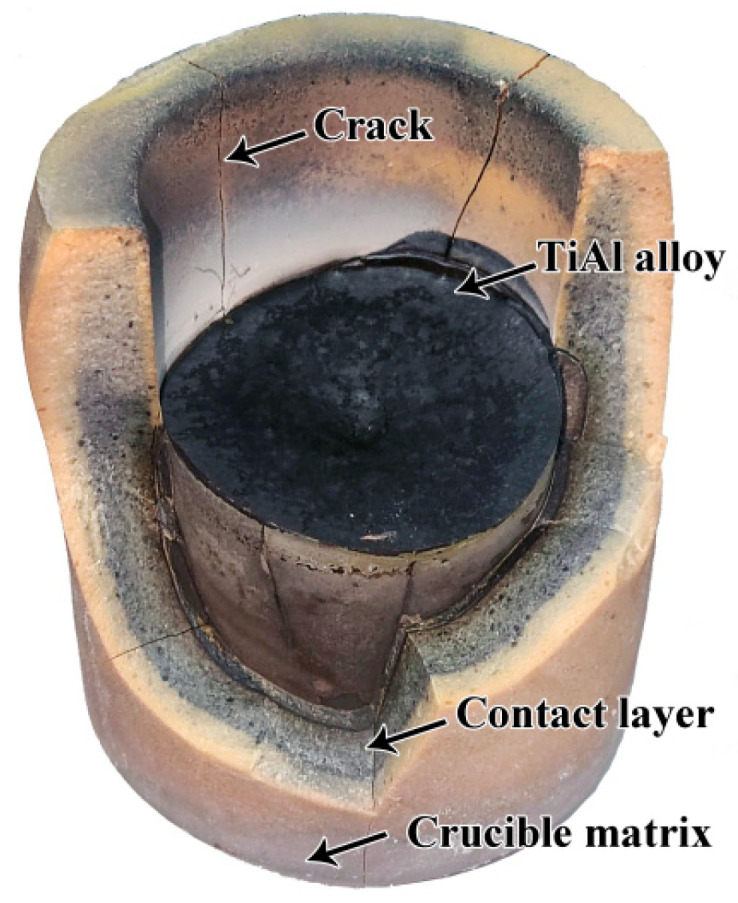
The macroscopic picture of the TiAl alloy cooled in Sr_2_CeZrO_6_ crucible.

**Figure 6 materials-16-07298-f006:**
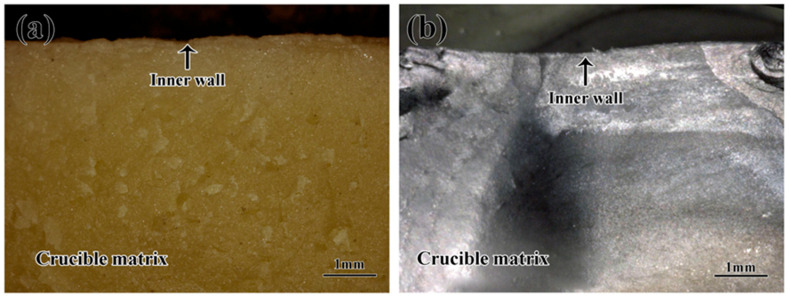
Macroscopic pictures of the crucibles: (**a**) before the melting, (**b**) after the melting.

**Figure 7 materials-16-07298-f007:**
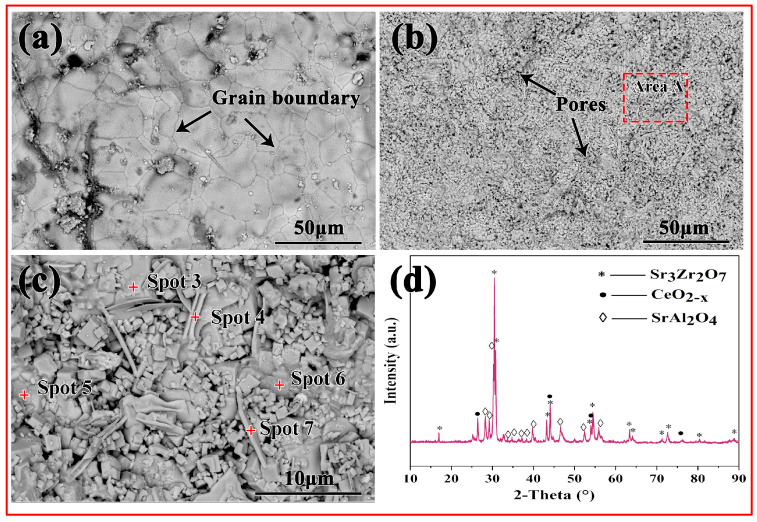
(**a**) SEM pictures of the crucible surface before melting; (**b**) SEM pictures of the crucible surface after melting; (**c**) the magnified picture of area A in (**b**); (**d**) XRD pattern of the crucible surface after melting.

**Figure 8 materials-16-07298-f008:**
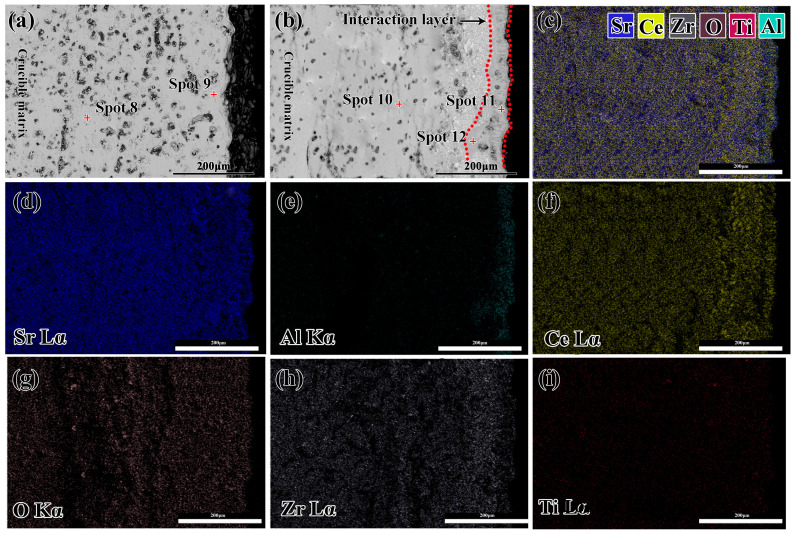
Map scanning image of the sidewall of the crucible before and after melting of the alloy; (**a**) the cross-section of the crucibles before melting; (**b**) the cross-section of the crucible after melting; (**c**) the combination of all elements in (**b**); (**d**–**i**) the EDS element mapping images for Sr, Ce, Zr, O, Zr and Ti.

**Figure 9 materials-16-07298-f009:**
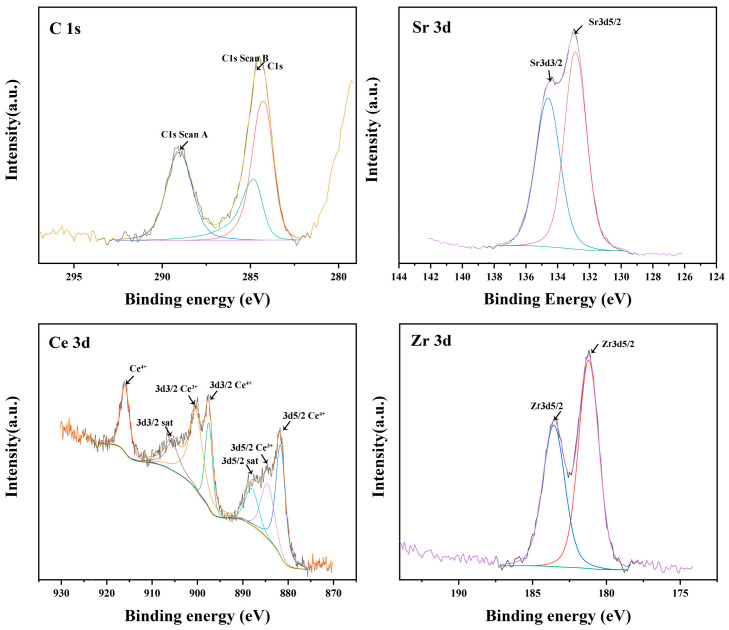
High-resolution X-ray photoelectron spectroscopy (XPS) spectra of C1s, Sr3d, Ce3d and Zr3d for the interaction layer.

**Figure 10 materials-16-07298-f010:**
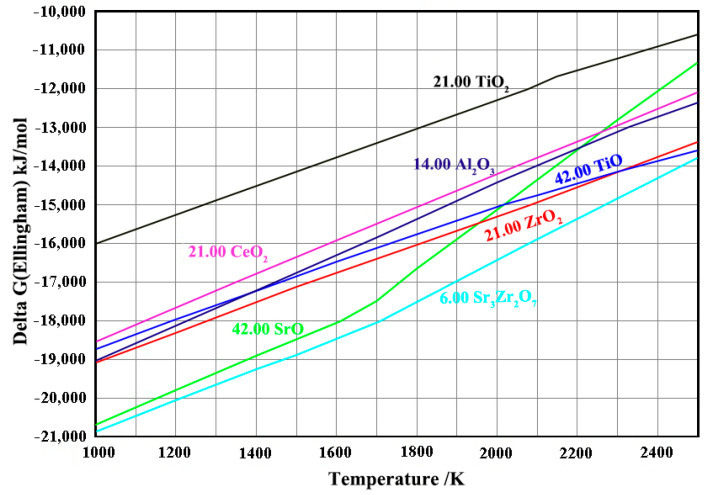
Gibbs free energy for the formation of TiO_2_, TiO, Al_2_O_3_, CeO, ZrO_2_, SrO and Sr_3_Zr_2_O_7_.

**Table 1 materials-16-07298-t001:** Refined structural parameters of Sr_2_CeZrO_6_ by Rietveld refinement of the orthorhombic structure in the Pnma space group (No. 62).

Atom	Wyckoff Position	x	y	z	Site Occ	Biso
Sr	4c	0.46569	0.25000	−0.00795	1.0000	0.1780(7)
Ce	4a	0.00000	0.00000	0.00000	0.5000	0.00805(2)
Zr	4a	0.00000	0.00000	0.00000	0.5000	0.00810(2)
O	4c	0.02400	0.25000	0.09200	1.00000	0.03900(2)
O	8d	0.02874	−0.04360	0.20730	1.00000	0.02960(3)

Crystal structure: orthorhombic; space group (No. 62): Pnma, a = 5.9742(3) Å, b = 8.3910(5) Å, c = 5.9069(5) Å, volume of unit cell = 296.11(6) Å3; Rwp = 8.94%, GOF = 1.57, bond lengths (Å): Sr–O = 2.9809(5) Å × 12, (Sr/Zr)–O = 2.8639(3) Å × 6.

**Table 2 materials-16-07298-t002:** EDS results of spots 1 and 2 in Figure 4a.

	Element at%			
	Sr	Ce	Zr	O
Spot 1	13.99	8.56	7.53	69.92
Spot 2	17.36	6.97	7.15	68.52

**Table 3 materials-16-07298-t003:** EDS results of spots 3–7 in Figure 7.

	Element at%					
	Sr	Ce	Zr	O	Al	Ti
Spot 3	19.50	9.30	1.95	40.12	26.02	3.11
Spot 4	13.84	13.94	2.61	57.14	11.99	0.48
Spot 5	24.34	1.32	16.07	55.03	3.12	0.12
Spot 6	10.04	6.97	2.05	65.72	14.10	1.12
Spot 7	11.28	7.81	2.16	62.75	13.84	2.16

**Table 4 materials-16-07298-t004:** EDS results of spots 8~12 in Figure 8.

	Element at%					
	Sr	Ce	Zr	O	Al	Ti
Spot 8	19.86	11.31	10.44	58.39	/	/
Spot 9	20.27	10.83	10.86	58.04	/	/
Spot 10	21.61	10.57	11.49	56.33	/	/
Spot 11	10.53	26.40	3.68	59.39	/	/
Spot 12	20.46	3.94	14.48	57.02	4.10	/

## Data Availability

Data are contained within the article.

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
