# Peer review of "Preparation of Sr2CeZrO6 Refractory and Its Interaction with TiAl Alloy"

_materials, 2023, doi:10.3390/ma16237298_

Round 1
Reviewer 1 Report
Comments and Suggestions for Authors
Title : Preparation of Sr2CeZrO6 refractory and its interaction with TiAl alloy
Review: The paper seems to deal with creating a new refractory to improvise TiAl alloy manufacturing process. Though the goal may be interesting but the paper needs much improvisation before acceptable in this journal. Key points for improvisation:
1. Why the introduction starts with Ti-Al alloy, thought the emphasis was on oxides, if it is otherwise abstract need to be modified similarly.
2. XRD has been shown but the XRD of all the constituent oxide need to be shown.
3. Why the specific ration of the constituent oxides are chosen need to be explained.
4. Control experiments need to be carried out to rationale the mechanism.
5. If control experiments is unavailable one should try using first principle calculations.
6. There is no XPS to show the nature of intercation between the TiAl and the oxide.
7. the SEM picture resolution to be improved a more visible picture for the spots to be shown.
8. One need to explain how EDS samples are taken for the spots
9. No phase diagrams are shown, nor the phase behavior is understood from the XRD.
10. We do not see a annealing study to see the impact of temperature cycle on the crucible.
Comments on the Quality of English Language
Moderate editing of English language required
Reviewer 2 Report
Comments and Suggestions for Authors
The manuscript "Preparation of Sr2CeZrO6refractory and its interaction with TiAl alloy" has been reviewed. It deals with a cold isostatic pressing and solid state synthesis of refractory and its interaction with TiAl alloy melt.
The experimental work is interesting and well organized with SEM, BSE, EDS and XRD experimental techniques. The interaction between alloy and crucible has been characterized.
The manuscript is clear, well organized and supported by the results. English is almost fine.
In my opinion it can be accepted after the following minor revisions:
X axis, fig. 2: Theta (not THEATA);
Fig. 6 d): font size must be bigger to increase readability;
Table 3: please check composition (line 1 sum is 110%, line 2 sum is 98,4 %);
Fig. 8. It is not clear where relationships DeltaG vs T are taken from. Please specify;
After that the manuscript can be accepted.
Comments on the Quality of English Language
English is almost fine
Reviewer 3 Report
Comments and Suggestions for Authors
This paper studies the production of strontium-cesium-zirconium-based refractory crucibles with a solid state reaction and investigates interactions with melt TiAl alloy in investment casting. This is a straightforward, good experimental paper. I have only few minor comments:
Introduction gives shortly some background information and lists some important references before the definition of the research problem. At the end of Introduction, the authors name barium carbonate as one raw material for the crucibles. I guess they mean strontium carbonate.
How reliable are the results? How many repetitions were carried out? The authors speak (in Section 2) about “crucibles”, but do not clearly tell, how extensive experimental material they had available.
Some panels in Fig. 7 are impossible to read. These panels (Fig. 7 (c-i) have not been commented too much in the text. Are they necessary at all?
Conclusions need some comments on the practical meaning of the results and a couple of sentences about the further research needed.
The language of the paper is mainly good and readable. I found some errors:
line 41: these refractory materials
54: materials such as Sr-Zr oxides exhibit
96-7: the crucible biscuits were
124: It can be seen
174: with high chemical activity
192-3: the grain boundaries had disappeared
206: From Fig.7a
229: to the high chemical activity
230: to the generation of
253-4: The products were generated
Comments on the Quality of English LanguageOK.
Reviewer 4 Report
Comments and Suggestions for Authors
In my opinion, the contents of this manuscript “Preparation of Sr2CeZrO6 refractory and its interaction with TiAl alloy” by F. Bian et al. add a significant contribution to the theory of the interaction of Sr2CeZrO6 refractory with TiAl alloy melt was investigated.
1) The authors should explain clearly in the abstract what is the novelty of the proposed method and what is the added value in this article and need to emphasize more clearly the contribution of the manuscript from a scientific point of view.
2) Conclusions of the study should be summarized. Please focus on the main achievements.
3) For the sake of completeness, the authors could also mention the following paper, in which the same topics or using similar procedures are discussed, for example: Effects of Laser Shock Peening on Microstructure and Properties of Ti–6Al–4V Titanium Alloy Fabricated via Selective Laser Melting, Materials 2020, 13(15), 3261; https://doi.org/10.3390/ma13153261, Mechanics of Elastic Composites Chapman and Hall/CRC, USA, (2004).
To summarize, I recommend the acceptance of this paper after Minor Revisions.
Round 2
Reviewer 1 Report
Comments and Suggestions for Authors
The author has updated according to my suggestions.
Comments on the Quality of English Languagethis looks good to me